# A novel class of polymorphic toxins in Bacteroidetes

Biswanath Jana[1], Dor Salomon[1] , Eran Bosis[2]

Bacteroidetes are Gram-negative bacteria that are abundant in the environment as well as in the gut microbiota of animals. Many bacteroidetes encode large proteins containing an N-terminal domain of unknown function, named TANFOR. In this work, we show that TANFOR-containing proteins carry polymorphic C-terminal toxin domains with predicted antibacterial and anti-eukaryotic activities. We also show that a C-terminal domain that is prevalent in TANFOR-containing proteins represents a novel family of antibacterial DNase toxins, which we named BaCT (Bacteroidetes C-terminal Toxin). Finally, we discover that TANFOR-encoding gene neighborhoods are enriched with genes that encode substrates of the type IX secretion system (T9SS), which is involved in exporting proteins from the periplasm across the outer membrane. Based on these findings, we conclude that TANFOR-containing proteins are a new class of polymorphic toxins, and we hypothesize that they are T9SS substrates.

## Introduction

Bacteria use specialized protein delivery systems to control their environment. Nine major secretion systems, named type I–IX secretion system (T1SS-T9SS), have been defined to date; their distribution varies between bacterial phyla ([1], [2], [3]). The proteins secreted by these systems are often effectors that possess toxic activities. Many of these effectors can be classified as polymorphic toxins, which are multi-domain proteins comprising a domain that dictates the trafficking mode and a toxin domain ([4], [5]). Toxin domains may mediate antibacterial or anti-eukaryotic activities, and therefore, such secreted polymorphic toxins play a role in bacterial competition or virulence, respectively. Whereas some secretion systems only target one kingdom of life (e.g., T3SS, which targets eukaryotes ([6])), others were found to function as trans-kingdom weapons (e.g., T4SS ([7], [8]) and T6SS ([9], [10]), which target both bacteria and eukaryotes).

In Gram-negative bacteria, secretion occurs as a one- or two-step process ([11]). In one-step secretion, the effector is directly translocated across both cell membranes. In two-step secretion, the effector is first translocated through the inner membrane to the periplasm by the general secretion (Sec) pathway or the twin-arginine pathway; then it is exported through the outer membrane via specialized secretion machinery.

The most recently discovered secretion system, T9SS ([2], [12], [13]), is restricted to members of the Bacteroidetes phylum ([3], [14]). It secretes proteins that have been shown to play a role in gliding motility and virulence ([2], [12], [15], [16], [17]). T9SS substrates are secreted in a two-step process. They contain an N-terminal signal peptide that mediates their translocation via the Sec system to the periplasm; a conserved C-terminal domain (named CTD or Por_Secre_tail) is then recognized in the periplasm by T9SS, which mediates export through the outer membrane ([16], [18], [19], [20]). Although many T9SS substrates were discovered by the presence of CTDs ([20]), it is possible that other mechanisms of secretion via T9SS exist, and thus, additional secreted substrates will be revealed.

In a recent study, we characterized a family of antibacterial DNase toxin domains, named PoNe (Polymorphic Nuclease effector), which were found in secreted effectors containing delivery domains of T5SS, T6SS, and T7SS ([21]). PoNe domains were also found fused to N-terminal domains not previously associated with specific secretion systems, suggesting that these N-terminal domains may be novel delivery domains that represent new classes of polymorphic toxins ([21]). One of these proteins, WP_026982765.1 encoded by *Flavobacterium* sp. URHB0058, contains a C-terminal PoNe domain fused to an N-terminal TANFOR domain (TIGR02542, also named T_forsyth_147). The TANFOR domain was previously defined as a region of ~147 residues found at the N-termini of large bacterial proteins, often accompanied by fibronectin type III (Fn3) domains.

In this work, we investigated the possibility that TANFOR-containing proteins are polymorphic toxins. We found that many TANFOR-containing proteins, which are almost exclusively restricted to members of the Bacteroidetes phylum, carry polymorphic C-terminal toxin domains with predicted antibacterial and anti-eukaryotic activities. We showed that a previously undefined C-terminal domain, named BaCT (Bacteroidetes C-terminal Toxin), that is prevalent in TANFOR-containing proteins represents a novel family of antibacterial DNase toxins. We also identified its cognate immunity protein family. Finally, we demonstrated T9SS-secreted substrates are enriched in the neighborhoods of TANFOR-encoding genes. Based on these findings, we propose that TANFOR-containing

[1]Department of Clinical Microbiology and Immunology, Sackler Faculty of Medicine, Tel Aviv University, Tel Aviv, Israel   [2]Department of Biotechnology Engineering, ORT Braude College of Engineering, Karmiel, Israel

Correspondence: dorsalomon@mail.tau.ac.il; bosis@braude.ac.il

proteins are a new class of polymorphic toxins. We also hypothesize that TANFOR-containing proteins represent a new class of T9SS substrates.

## Results and Discussion

### Proteins containing a TANFOR domain are polymorphic toxins

We hypothesized that TANFOR domain–containing proteins can carry polymorphic toxin domains other than PoNe. To test this, we used reverse position-specific BLAST (RPS-BLAST) (22) and identified 339 TANFOR-containing proteins that were encoded almost exclusively by members of the Bacteroidetes phylum (Supplemental Data 1). We noted that 5.8% of available RefSeq Bactroidetes genomes encode at least one, and up to five TANFOR-containing proteins (Supplemental Data 2). Analysis of domains fused to the N-terminal TANFOR domain revealed many known and predicted C-terminal toxin domains (Fig 1 and Supplemental Data 1), including antibacterial nucleases (e.g., AHH, WHH, NUC, EndoU, and Tox-REase), peptidoglycan hydrolases (e.g., Lyz_endolysin_autolysin), and even predicted anti-eukaryotic toxins (e.g., the actin ADP-ribosylating toxin VIP2). The predicted antibacterial toxins were often accompanied by downstream-encoded known (e.g., Imm49, Imm74, and SUKH-1) or predicted immunity proteins. Taking the above results together, we propose that TANFOR-containing proteins are a new class of polymorphic toxins with antibacterial and anti-eukaryotic activities.

Polymorphic toxins are often secreted proteins that are used by bacteria to mediate interbacterial conflicts or interactions with eukaryotic hosts (4, 9, 21, 23, 24, 25, 26, 27). Notably, ~87% of the identified TANFOR-containing proteins were predicted to harbor an N-terminal Sec secretion signal (by either the SignalP 5.0 (28) or Phobius (29) prediction servers), suggesting that TANFOR-containing proteins are transported to the periplasm. Because anti-eukaryotic and nuclease toxin domains, such as those identified in TANFOR-containing proteins, are not likely to function in the bacterial periplasm, we hypothesize that they may be exported out of the cell.

Interestingly, we found that in several instances, a predicted toxin domain was encoded by the gene immediately downstream of the TANFOR-encoding gene, rather than being part of the TANFOR-containing protein (Fig 1 and Supplemental Data 1). When the downstream-encoded toxin was predicted to mediate antibacterial activity, we also found an adjacent gene encoding a predicted immunity protein. A similar phenomenon, in which N-terminal secretion or delivery domains either carry the toxin domain in the same polypeptide chain or are encoded by an adjacent gene, was previously observed in T6SS effectors containing delivery domains such as MIX and FIX (21, 30). It is possible that in such cases when the toxin is encoded by a separate gene, the protein product of the toxin gene interacts with the adjacently encoded TANFOR-containing protein.

### BaCT is a novel DNase toxin domain

Many TANFOR-containing proteins did not contain a predicted toxin domain and did not have an identifiable toxin encoded

**Figure 1. Many TANFOR-containing proteins carry C-terminal toxin domains.** The domain architecture of TANFOR-containing proteins and downstream encoded proteins (Downstream 1, proteins encoded immediately downstream of TANFOR-encoding gene; Downstream 2, proteins encoded by the second open reading frame downstream of TANFOR-encoding genes). In TANFOR-containing proteins, only the N-terminal TANFOR and the C-terminal domain are shown. The dashed lines denote protein regions that may contain additional domains, which are not shown. Domain sizes are not to scale. "No. of occurrences" refers to the number of TANFOR-containing proteins that were found to have the indicated domain architecture. Known or predicted immunity (Putative_Imm) proteins are also shown and are denoted in cyan. Undefined domains are denoted as "Unknown." AHH and WHH, nucleases of the HNH/Endo VII superfamily; BaCT, Bacteroidetes C-terminal Toxin; BaCTic, BaCT immunity component; GIY-YIG, endoribonuclease containing a GIY-YIG motif; EndoU, endonuclease; Imm49, immunity protein 49; Imm74, immunity protein 74; NUC, nuclease; PoNe, polymorphic nuclease effector; PoNi, PoNe immunity; SUKH-1 and SUKH_6, Syd US22 Knr4 homology superfamily; Tnp, transposase; TNT, tuberculosis necrotizing toxin; Tox-deaminase, toxin deaminase; Tox-REase-5, restriction endonuclease fold toxin 5; VIP2, vegetative insecticidal protein 2; YeeF, predicted ribonuclease toxin component of the YeeF-YezG toxin–antitoxin module.

downstream (Fig 1). Therefore, we set out to determine whether the C-termini of TANFOR-containing proteins with no defined activity are novel toxin domains. Analysis of the C-terminal sequences of TANFOR-containing proteins revealed a new domain, which we named BaCT (Bacteroidetes C-terminal Toxin) (Fig 1 and

Supplemental Data 1). BaCT was found at the C-termini of 18 TANFOR-containing proteins and 7 proteins encoded immediately downstream of TANFOR-encoding genes. We hypothesized that BaCT functions as an antibacterial toxin domain. Indeed, inducible expression of the BaCT domain of WP_081912444.1 (302 C-terminal amino acids), a TANFOR-containing protein of *Flavobacterium hydatis* DSM 2063, was deleterious to *Escherichia coli* (Fig 2A).

We used RPS-BLAST to search genomes that are available in the RefSeq database for BaCT homologs. We identified 97 proteins containing BaCT, all but four of which were found in members of the Bacteroidetes phylum (Fig 1 and Supplemental Data 3). BaCT domains were fused to N-terminal TANFOR, Fn3, PT-HINT, or DUF4157 domains (Fig 2B). The latter two have been previously found in secreted polymorphic toxins (4, 21, 31, 32). Furthermore, multiple sequence alignment of the identified BaCT domains revealed a conserved region centered around an invariant (D/E)xK (Figs 2C and S1). We noted that this conserved region, when superimposed onto predicted secondary structure elements, was similar to the canonical motif shared by members of the PD-(D/E)xK phosphodiesterase superfamily (33, 34) (Figs 2C and S1); many members of this superfamily are DNases (21, 34). Therefore, we hypothesized that BaCT represents a new family within this highly divergent superfamily, and that it has DNase activity. In agreement with this hypothesis, we found that the purified BaCT domain of WP_081912444.1 (302 C-terminal amino acids) degraded the genomic DNA of *E. coli* in vitro, in a magnesium-dependent manner (Figs 2D and S2). BaCT also induced DNA degradation in vivo (Fig 2E), as well as the corresponding activation of a fluorescent SOS response reporter (Fig 2F), indicative of DNA damage (35).

Consistent with our prediction, mutations in residues of the conserved BaCT ExK motif to alanine (E1724A and K1726) resulted in loss of toxicity upon expression in *E. coli* (Fig 2A), as well as loss of DNase activity in vitro and in vivo (Fig 2D–F). The mutant forms of BaCT were expressed, as evident by immunoblotting (Fig 2G), demonstrating that the loss of toxicity did not result from impaired expression of the mutants. Taken together, the above results indicate that the BaCT domains are DNases that represent a new toxin family belonging to the PD-(D/E)xK phosphodiesterase superfamily. Moreover, our findings suggest that additional toxin domains can be revealed by investigating C-terminal regions of TANFOR-containing proteins in which no predicted toxin domain was identified.

### The gene downstream of BaCT encodes its cognate immunity protein

The above results indicated that the BaCT-containing WP_081912444.1 is an antibacterial toxin. Therefore, we predicted that the downstream-encoded protein, WP_035620349.1, functions as the cognate immunity protein of BaCT and provides protection against self-intoxication. Indeed, co-expression of the downstream-encoded protein antagonized the BaCT-mediated toxicity in *E. coli*, but not the toxicity of PoNe[Vp], a member of a different family of antibacterial PD-(D/E)xK phosphodiesterases (21) (Fig 3A). Moreover, we found that WP_035620349.1 directly interacts with wild-type and mutated forms of BaCT in a pull-down assay (Fig 3B). Based on these results, we conclude that WP_035620349.1 is the cognate immunity protein of BaCT, and that it likely antagonizes BaCT toxicity via direct binding to the toxin domain.

Next, we set out to characterize the BaCT immunity protein family. The sequences of proteins encoded immediately downstream of BaCT family members (within 50 bp, on the same DNA strand), which are predicted to be their cognate immunity proteins, were considerably diverse and could not be presented as a conserved domain using traditional multiple sequence alignment tools. To determine whether these proteins share some degree of sequence similarity, we clustered them using all-against-all sequence pairwise similarity, which was performed using the CLANS classification tool (36). This analysis revealed that 57 out of 65 protein sequences (87.7%) were grouped together (Fig 3C), suggesting that they belong to the same protein family. Therefore, we named them BaCTic (BaCT immunity component). It is possible that the remaining proteins encoded downstream of BaCT family members either neighbor-inactive BaCT domains or constitute distinct immunity families. A similar phenomenon in which multiple immunity families were found to antagonize one toxin family had been previously described for Colicin M, a family of antibacterial, peptidoglycan-targeting toxins (37).

### T9SS-secreted proteins are enriched in TANFOR-encoding gene neighborhoods

While analyzing the genome neighborhood of the gene encoding the abovementioned TANFOR- and BaCT-containing protein (WP_081912444.1), we noted that this toxin was encoded immediately downstream of a gene encoding a predicted T9SS-secreted substrate containing SprB repeats and a CTD (Fig 4 and Supplemental Data 1). Interestingly, we found similar predicted T9SS substrates encoded in the vicinity of other TANFOR-encoding genes (Fig 4). These observations prompted us to investigate which domains are enriched in genome neighborhoods of TANFOR-encoding genes because such genetic interactions can imply functional association. To this end, we identified the domains found within 10 upstream and 10 downstream genes of each TANFOR-encoding gene. For each conserved domain that was identified, we determined the occurrence of this domain in TANFOR-encoding gene neighborhoods. The same analysis was performed for a group of randomly sampled, non-TANFOR-encoding genes. For each identified domain, we then determined whether it is enriched in TANFOR-encoding gene neighborhoods (Supplemental Data 4). Interestingly, two of the enriched domains were the abovementioned SprB and the T9SS export signal CTD (Por_Secre_tail) (Table 1), which are directly associated with T9SS substrates. The additional enriched domains, Fn3 and polycystic kidney disease (shared between COG3291, PCC, and polycystic kidney disease in Table 1), are not associated with any specific cellular activity or protein secretion system. Remarkably, we also found that >95% of the genomes encoding a TANFOR domain harbor T9SS (Supplemental Data 2 and 5). These findings suggest a possible correlation between polymorphic toxins of the TANFOR-containing class and T9SS substrates. Based on these results, in conjunction with our observation that most TANFOR-containing toxins have an N-terminal periplasmic signal peptide (Supplemental Data 1), it is interesting to speculate that TANFOR-containing toxins are exported by T9SS, the only two-step protein secretion system that is widespread in Bacteroidetes (3). If true, this would imply a role for T9SS in interbacterial competition. Future work is required to establish

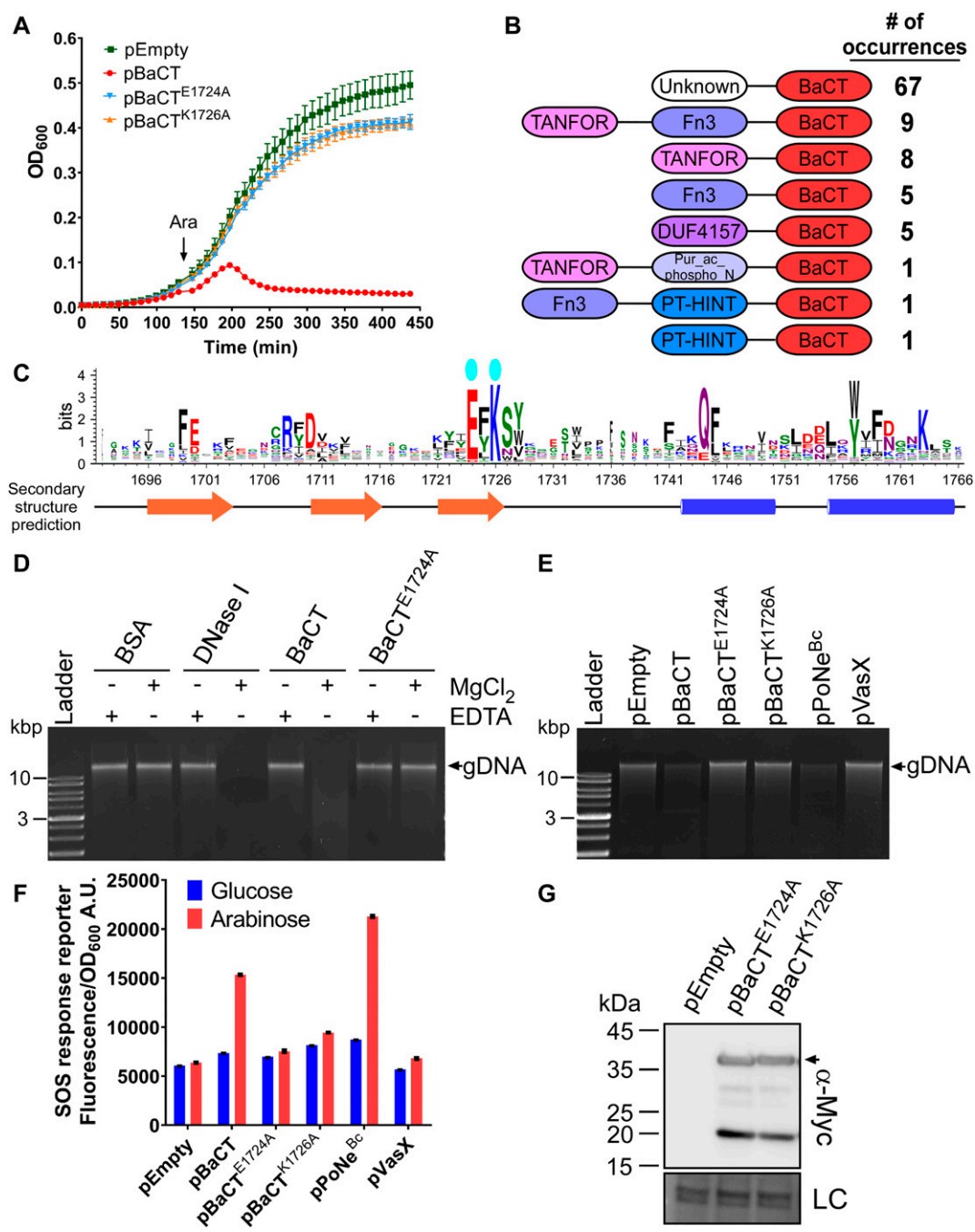

**Figure 2. BaCT is a novel antibacterial DNase domain.**
**(A)** The toxicity of BaCT in bacteria. The growth of *E. coli* BL21 (DE3) containing arabinose-inducible plasmids for the expression of the indicated BaCT variants. The arrow denotes the time at which L-arabinose was added to the media. **(B)** The domain architecture of BaCT-containing proteins. "No. of occurrences" refers to the number of BaCT-containing proteins that were found to have the indicated domain architecture. **(C)** A conserved motif found in BaCT proteins is illustrated using WebLogo 3, based on multiple sequence alignment of homologs of the 300 C-terminal residues in WP_081912444.1. Cyan ovals denote conserved predicted catalytic residues. Secondary structure prediction (by Jpred) is provided below. Alpha helices are denoted by blue cylinders, and beta strands by orange arrows. **(D)** An in vitro DNase activity assay. Purified *E. coli* genomic DNA (gDNA) was co-incubated with the indicated purified proteins in the presence (+) or absence (−) of $Mg^{2+}$ or EDTA at 37°C for 10 min. BSA and DNase I were used as negative and positive controls, respectively. **(E)** An in vivo DNase activity assay. Genomic DNA (gDNA) was extracted from 1.0 $OD_{600}$ units of *E. coli* BL21 (DE3) expressing the indicated proteins from arabinose-inducible expression plasmids. The *Bacillus cereus* PoNe DNase toxin BC3021 (pPoNe[Bc]) was used as the positive control, and the pore-forming *Vibrio cholerae* T6SS effector VasX, fused to an N-terminal PelB Sec secretion signal (pVasX), was used as the negative control. **(F)** SOS response activation in *E. coli* BL21 (DE3) harboring the indicated arabinose-inducible expression plasmids. SOS response levels were measured as GFP fluorescence from the reporter plasmid pL(*lexO*)-GFP, divided by cell density ($OD_{600}$) and are shown as a.u. Data represent the mean ± SD (n = 3). **(G)** The expression of the indicated C-terminally myc-tagged forms of BaCT from arabinose-inducible plasmids in *E. coli* BL21 (DE3). Proteins were detected by immunoblotting using specific antibodies against myc. Loading control, visualized as trihalo compounds' fluorescence of the immunoblot membrane (Stain-Free imaging technology; Bio-Rad), is shown for total protein lysates. An arrow denotes the expected protein size.

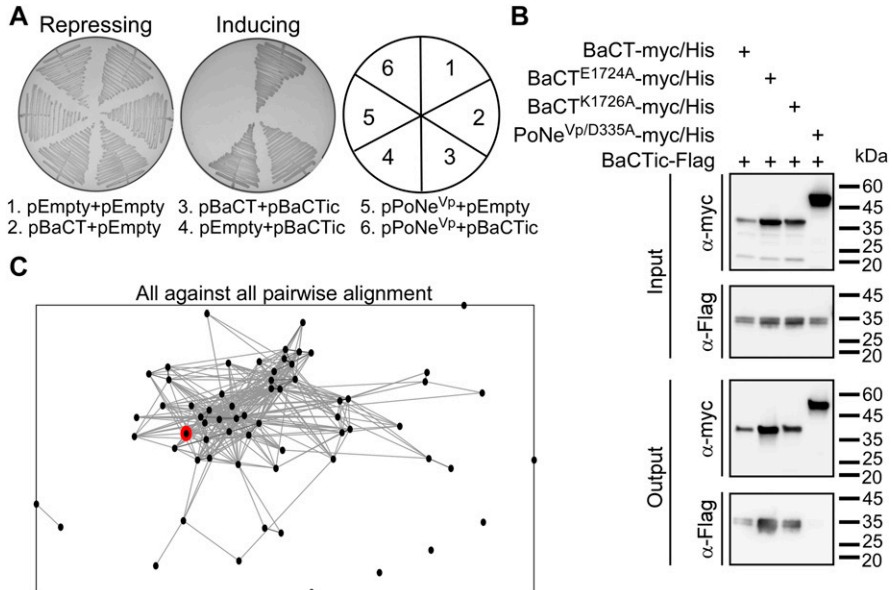

**Figure 3. BaCTic provides immunity against BaCT-mediated toxicity.**
**(A)** Rescue of BaCT-mediated toxicity by cognate BaCTic. Toxicity of BaCT expressed in *E. coli* BL21 (DE3) from an arabinose-inducible expression plasmid with or without BaCTic (WP_035620349.1). The *Vibrio parahaemolyticus* PoNe DNase toxin B5C30_RS14465 (pPoNe$^{Vp}$) was used as a control. Repressing, 0.4% glucose; inducing, 0.02% arabinose. **(B)** BaCTic specifically binds BaCT. Ni-Sepharose resin pull-down of myc/His–tagged BaCT variants or PoNe$^{Vp/D335A}$ (used as a control) co-expressed with Flag-tagged BaCTic in *E. coli* BL21 (DE3). **(C)** Proteins encoded immediately downstream (within 50 bp) of BaCT-encoding genes (i.e., BaCTic) were clustered in two dimensions based on all-against-all sequence similarity (using CLANS). Circles denote individual proteins and connecting lines denote sequence similarity. WP_035620349.1 is denoted in red.

an experimental model system that will enable investigating these hypotheses.

# Materials and Methods

### Strains and media

*E. coli* strain DH5α (λ *pir*) (a gift from Eric V Stabb) was used for plasmid maintenance and amplification. *E. coli* strain BL21 (DE3) was used for protein expression and for toxicity and SOS response assays. *E. coli* were routinely grown in 2×YT broth (1.6% wt/vol tryptone, 1% wt/vol yeast extract, and 0.5% wt/vol NaCl) or on Lysogeny broth (LB, 1% wt/vol NaCl) agar plates at 37°C. When necessary to maintain plasmids, media were supplemented with 10 $\mu$g/ml chloramphenicol, 100 $\mu$g/ml ampicillin, or 30 $\mu$g/ml kanamycin. To induce the expression of genes from a plasmid, 0.02–0.1% (wt/vol) L-arabinose was included in the media, as indicated.

### Plasmid construction and mutagenesis

For arabinose-inducible expression of BaCT, the region corresponding to the 302 C-terminal residues of the gene encoding WP_081912444.1 was amplified from the genomic DNA of *F. hydatis* DSM 2063 (purchased from the DSMZ collection). PCR fragments were inserted into the multiple cloning site (MCS) of the pBAD/Myc–His vector (Invitrogen) harboring a kanamycin-resistance cassette (38), in-frame with the C-terminal Myc-6xHis tag, to produce pBaCT. To generate BaCT$^{E1724A}$ and BaCT$^{K1726A}$, site-directed mutagenesis was performed using the abovementioned pBAD/Myc–His plasmid as a template. The construction of pPoNe$^{Vp}$ (B5C30_RS14465) and pPoNe$^{Bc}$ (BC3021) was reported previously (21). For the expression of periplasmic VasX (WP_000070352.1), the corresponding coding sequence from *Vibrio cholerae* V52 was

inserted into the Kan$^R$pPER5/*Myc*–His vector (30), a pBAD derivative in which the PelB signal sequence was inserted at the 5' of the MCS.

For arabinose-inducible expression of BaCTic, the coding sequence of WP_035620349.1 was amplified with an in-frame C-terminal Flag tag from the genomic DNA of *F. hydatis* DSM 2063. PCR fragments were inserted into the MCS of pBAD33.1 (Addgene) to produce pBaCTic. All constructs were confirmed by DNA sequencing.

### Bacterial toxicity assays

To assess the toxicity of BaCT, *E. coli* BL21 (DE3) were transformed with the indicated empty plasmid (pEmpty) or pBaCT arabinose-inducible expression plasmids (pBaCT). Transformants were grown overnight in 2×YT supplemented with kanamycin and 0.2% (wt/vol) glucose (to repress leaky expression from the *Pbad* promoter). Cultures were washed to remove residual glucose and normalized to OD$_{600}$ = 1 in 2×YT. Next, the cultures were further diluted to OD$_{600}$ = 0.01 in 2×YT containing kanamycin (30 $\mu$g/ml), and 200 $\mu$l of each sample were transferred into 96-well plates in triplicate. OD$_{600}$ readings were taken every 10 min for 7 h while plates were grown at 37°C with agitation (205 cpm) in a microplate reader (BioTek SYNERGY H1). After 2 h of growth, L-arabinose was added to each well to a final concentration of 0.1% (wt/vol), to induce expression from the plasmids. The experiment was repeated three times with similar results, and the results from a representative experiment are shown.

To assess the protection conferred by BaCTic against BaCT-mediated toxicity, arabinose-inducible pBaCTic or an empty pBAD33.1 expression plasmid was co-transformed with the pBAD toxin-expression plasmids (pBaCT or pPoNe) into *E. coli* BL21 (DE3). Transformants were streaked onto either repressing (containing 0.4% wt/vol glucose) or inducing (containing 0.02% wt/vol L-arabinose) LB agar plates supplemented with kanamycin (30 $\mu$g/ml) and chloramphenicol (10 $\mu$g/ml). The plates were incubated at 30°C overnight

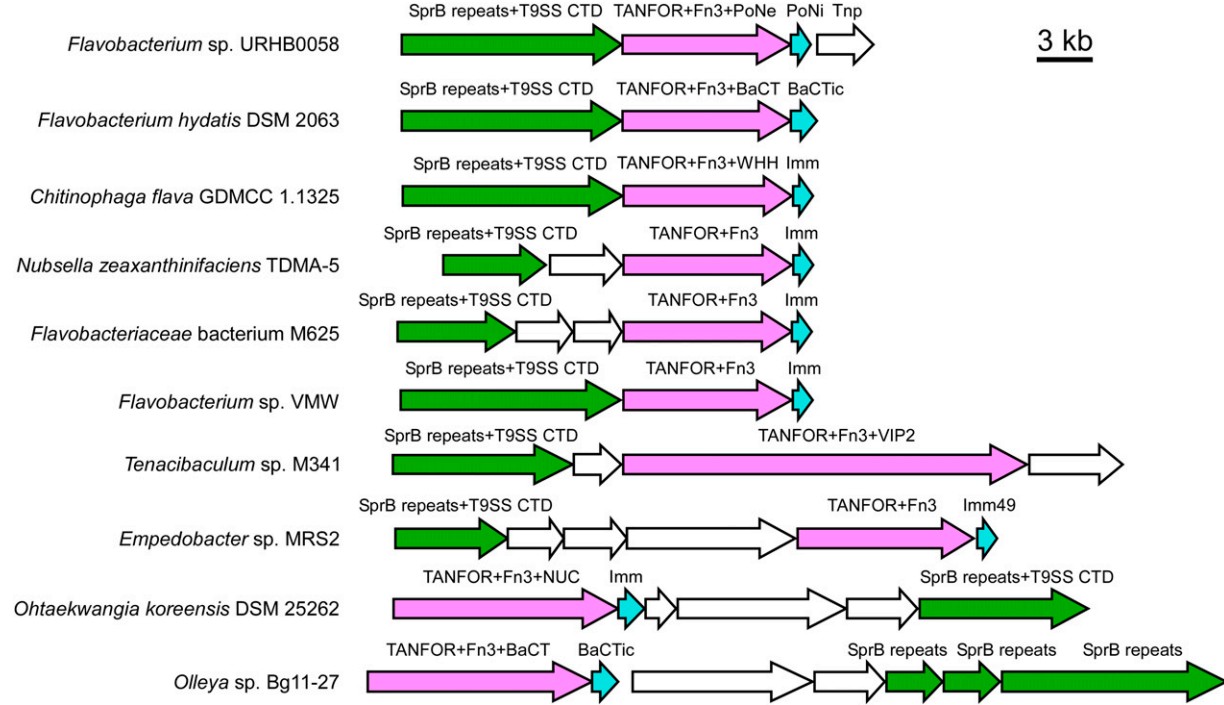

**Figure 4. TANFOR-encoding genes often neighbor T9SS-secreted proteins.**
Gene neighborhoods of selected TANFOR-encoding genes are shown. Genes are represented by arrows indicating the direction of translation. Encoded domains are shown above. Organism names are shown on the left. TANFOR-encoding genes are denoted in pink, known or predicted immunity genes are denoted in cyan, and T9SS-associated genes are denoted in green. CTD, T9SS C-terminal domain; Fn3, fibronectin type III; Imm, immunity.

before they were photographed. The experiment was repeated three times with similar results, and the results from a representative experiment are shown.

### Protein expression in *E. coli*

To verify the expression of mutant BaCT proteins in *E. coli*, the cultures from the bacterial toxicity assay (described above) were collected from the 96-well plates 5 h post-induction. The cells were pelleted and resuspended in a (2×) tris-glycine SDS sample buffer (Novex), followed by boiling at 95°C for 10 min. Samples were resolved on TGX stain-free gels (Bio-Rad), and then transferred onto

nitrocellulose membranes. Immunoblotting was performed with anti-c-Myc antibodies (9E10, sc-40; Santa Cruz Biotechnology) at 1:1,000 dilution. Protein signals were visualized by ECL.

### In vitro DNase assays

Overnight cultures of *E. coli* BL21 (DE3) containing plasmids for the arabinose-inducible expression of the indicated Myc-6xHis–tagged BaCT variants and the Flag-tagged BaCTic were diluted 100-fold into 100 ml 2×YT media supplemented with kanamycin and chloramphenicol. Diluted cultures were incubated at 37°C with agitation (215 rpm). Protein expression was induced by adding 0.1% (wt/vol) L-arabinose when cultures reached an $OD_{600}$ of ~1.0, followed by

**Table 1. Domains enriched in TANFOR-encoding gene neighborhoods.**

| Domain | Description | *P*-value |
|---|---|---|
| pfam13573 | SprB repeat | 2.14E-87 |
| COG3401 | Fibronectin type 3 (Fn3) domain | 1.03E-56 |
| COG3291 | PKD repeat | 8.18E-47 |
| TIGR04183 | Por secretion system C-terminal sorting domain (Por_Secre_tail) | 1.63E-28 |
| TIGR00864 | Polycystin cation channel (PCC) protein containing PKD repeats | 2.04E-16 |
| cd00063 | Fibronectin type 3 (Fn3) domain; one of three types of internal repeats found in the plasma protein fibronectin | 5.87E-08 |
| cd00146 | Polycystic kidney disease I (PKD) domain, similar to other cell surface modules, with an IG-like fold | 2.16E-07 |

4-h incubation at 30°C with agitation (215 rpm). The cells were harvested by centrifugation and then resuspended in 5 ml lysis buffer A (20 mM Tris-Cl, pH 7.5, 500 mM NaCl, 5% vol/vol glycerol, 10 mM immidazole, 1 mM PMSF, and 6 M urea). Urea was used to denature the proteins so that BaCTic can be released from BaCT (see the pull-down assay below). The cells were disrupted using a high-pressure cell disruptor (Constant Systems One-Shot cell disruptor, model code: MC/AA). Cell debris was removed by centrifugation for 20 min at 17,000$g$ at 4°C. Supernatant fractions of lysed cells containing the denatured His-tagged BaCT variants were mixed with 200 $\mu$l Ni-Sepharose resin (50% slurry; GE Healthcare) and incubated for 1 h at 4°C with constant shaking. The suspensions (cell lysate with Ni-Sepharose resin) were loaded onto a column. A stepwise on-column refolding procedure was then applied. Immobilized resin was first washed with 2 ml wash buffer A1 (20 mM Tris-Cl, pH 7.5, 500 mM NaCl, 5% vol/vol glycerol, 40 mM imidazole, and 3 M urea), followed by washing with 8 ml wash buffer A2 (20 mM Tris-Cl, pH 7.5, 500 mM NaCl, 5% vol/vol glycerol, and 40 mM imidazole). Refolded proteins were eluted from the column using 1 ml of elution buffer A (20 mM Tris-Cl, pH 7.5, 500 mM NaCl, 5% vol/vol glycerol, and 500 mM imidazole). Buffers were kept at 4°C. The presence and purity of the eluted proteins were confirmed by SDS–PAGE. To exchange the buffer, purified proteins were dialyzed overnight against dialysis buffer (20 mM Tris-Cl, pH 7.5, 250 mM NaCl, and 5% vol/vol glycerol). The purified proteins were quantified by the Bradford method using 5× Bradford reagent (Bio-Rad).

For in vitro DNase activity, *E. coli* BL21 (DE3)–isolated genomic DNA (250 ng) was incubated with 0.25 $\mu$g of purified BaCT variants (as explained above) for 10 min at 37°C in DNase assay buffer (20 mM Tris-Cl, pH 7.5, 94 mM NaCl, and 2% vol/vol glycerol) supplemented with either 2 mM EDTA or 2 mM MgCl$_2$. The reactions were stopped by adding 6.65 $\mu$g of Proteinase K and incubating them at 55°C for 10 min. The samples were analyzed on 1.0% agarose gel electrophoresis. For positive and negative controls, 1 U DNase I (Thermo Fisher Scientific) and 0.25 $\mu$g BSA (Sigma-Aldrich) were used, respectively. The experiment was repeated three times with similar results, and the results from a representative experiment are shown.

### In vivo DNase assays

*E. coli* BL21 (DE3) containing the indicated pBAD/Myc–His plasmids for the expression of BaCT variants, PoNe[Bc], or VasX were grown overnight in 2×YT media supplemented with kanamycin and 0.2% (wt/vol) glucose. Overnight cultures were washed with 2×YT and normalized to an OD$_{600}$ of 1.0 in 3 ml of fresh 2×YT supplemented with kanamycin and 0.1% (wt/vol) L-arabinose (to induce protein expression). Cultures were grown for 90 additional minutes with agitation (215 rpm) at 37°C before 1.0 OD$_{600}$ units were pelleted. Genomic DNA was isolated from each sample using the EZ Spin Column Genomic DNA Kit (Bio Basic) and eluted with 30 $\mu$l of sterile distilled water (Milli-Q). Equal genomic DNA elution volumes were analyzed by 1.0% (wt/vol) agarose gel electrophoresis. The experiment was repeated three times with similar results. Results from a representative experiment are shown.

### SOS response activity

*E. coli* BL21(DE3) were co-transformed with the SOS response reporter plasmid pL(*lexO*)-GFP (39) and the indicated arabinose-inducible effector expression plasmid. Overnight cultures that were grown in 2×YT broth containing the appropriate antibiotics and 0.2% (wt/vol) glucose (to repress expression from *Pbad* promoter) were washed with fresh 2×YT. Washed suspensions were normalized to OD$_{600}$ = 0.5 in 900 $\mu$l of 2×YT supplemented with appropriate antibiotics and either 0.1% (wt/vol) glucose or 0.1% (wt/vol) L-arabinose to repress or induce gene expression, respectively. The cultures were grown at 37°C for 1 h and 30 min with agitation (215 rpm), before the cells were pelleted and washed with 1 ml of M9 media. The cells were then resuspended in 800 $\mu$l of M9 media, and 200 $\mu$l of cell suspension was transferred, in triplicate, into wells of a black 96-well plate with a clear optical bottom. GFP fluorescence (excitation 479 nm and emission 520 nm) and OD$_{600}$ were measured using a BioTek SYNERGY H1 microplate reader. Data are presented as arbitrary units resulting from the division of fluorescence measurements by OD$_{600}$ per well. The experiment was repeated three times with similar results. Results from a representative experiment are shown.

### Pull-down assays

Overnight cultures of *E. coli* BL21 (DE3) containing plasmids for the arabinose-inducible expression of the indicated Myc-6xHis–tagged PoNe[Vp/D335A] or BaCT variants and the Flag-tagged BaCTic were diluted 100-fold into 25 ml 2×YT media supplemented with kanamycin and chloramphenicol and incubated at 37°C with agitation (215 rpm). Protein expression was induced by adding 0.1% (wt/vol) L-arabinose when cultures reached an OD$_{600}$ of ~1.0, followed by incubation at 30°C for 4 h with agitation (215 rpm). Equal amounts of cells (40 OD$_{600}$ units) were harvested by centrifugation and then resuspended in 3 ml lysis buffer B (20 mM Tris-Cl, pH 7.5, 500 mM NaCl, 5% vol/vol glycerol, 10 mM immidazole, and 1 mM PMSF). The cells were disrupted using a high-pressure cell disruptor (Constant Systems One-Shot cell disruptor, model code: MC/AA). Cell debris was removed by centrifugation for 20 min at 17,000$g$ at 4°C. Next, 50 $\mu$l of supernatant fractions of lysed cells were taken and mixed with 50 $\mu$l of 2× protein sample buffer, boiled at 95°C for 5 min, and kept for subsequent input protein analysis. Then, the remaining supernatant fractions of lysed cells were mixed with 100 $\mu$l Ni-Sepharose resin (50% slurry; GE Healthcare) and incubated for 1 h at 4°C with constant shaking. Resin was collected by centrifugation at 2,000$g$ at 4°C, and washed with 10 ml wash buffer B (20 mM Tris-Cl, pH 7.5, 500 mM NaCl, 5% vol/vol glycerol, and 40 mM imidazole). Bound proteins were eluted from the resin using 200 $\mu$l of elution buffer B (20 mM Tris-Cl, pH 7.5, 500 mM NaCl, 5% vol/vol glycerol, and 500 mM imidazole). Buffers were kept at 4°C.

The eluted fractions (output) were mixed with 200 $\mu$l 2× protein sample buffer and boiled at 95°C for 5 min. The samples were resolved on TGX stain-free gels (Bio-Rad) and transferred onto PVDF membranes. The presence of PoNe or BaCT variants was detected using c-Myc (9E10) antibodies (sc-40; Santa Cruz Biotechnology), and the presence of BaCTic was detected using Flag

antibodies (DYKDDDDK Tag (D6W5B) Rabbit mAb #14793; Cell Signaling Technology) at 1:1,000 dilution. Protein signals were visualized by ECL. The experiment was repeated at least three times with similar results, and the results from a representative experiment are shown.

### Clustering of proteins encoded downstream of BaCT using CLANS

Downstream proteins encoded within 50 bp of identified BaCT-encoding genes were clustered in two dimensions based on all-against-all pairwise similarity using CLANS (36).

### Identification of TANFOR- and BaCT-containing proteins

The Position-Specific Scoring Matrix (PSSM) of the TANFOR domain (TIGR02542) was downloaded from the NCBI Conserved Domain Database (CDD) (40) on November 1, 2019. A local database containing all RefSeq genomes from NCBI was constructed (last updated on October 31, 2019). The PSSM of the TANFOR domain was used to identify bacterial genomes that contained the TANFOR domain. Subsequently, the protein sequences and feature tables of TANFOR-containing genomes were retrieved from the local database.

RPS-BLAST (22) was used to identify TANFOR-containing proteins. The results were filtered using an expected value threshold of $10^{-7}$ and a minimal alignment length of at least 90 aa. TANFOR-containing proteins were analyzed as described previously (21). Briefly, upstream and downstream proteins were identified, sequences were analyzed using the CDD (see below) to identify conserved domains (40), and transmembrane topology and signal peptides were predicted using Phobius (29) and SignalP 5.0 (28).

The PSSM of the BaCT domain was constructed using amino acids 1,542–1,841 of WP_081912444.1 from *F. hydatis* DSM 2063. Five iterations of PSI-BLAST (22) were performed against the reference protein database (a maximum of 500 hits with an expected value threshold of $10^{-6}$ were used in each iteration). Next, BaCT-containing genomes were retrieved and BaCT-containing proteins were identified. RPS-BLAST results were filtered using an expected value threshold of $10^{-9}$.

### Identification of conserved domains

The CDD version 3.17 and all related information were downloaded from NCBI on October 2, 2019. To identify conserved domains in protein sequences, RPS-BLAST was used and the output was processed using the Post-RPS-BLAST Processing Utility v0.1 available from the NCBI FTP server. The expect value threshold was set to $10^{-5}$.

### Illustration of the conserved residues and the predicted secondary structure of BaCT and PD-(D/E)xK families

BaCT domain sequences were aligned using Clustal Omega. Aligned columns not found in the *F. hydatis* DSM 2063 BaCT (WP_081912444.1; 300 C-terminal residues) were discarded. The BaCT domain conserved residues were illustrated using the WebLogo 3 server (41)

(http://weblogo.threeplusone.com). Multiple sequence alignment was visualized using JalView (42).

The secondary structure of the BaCT conserved region was predicted for the 300 C-terminal residues of WP_081912444.1 using the Jpred 4 server (43). Canonical secondary structures and conserved residues of PD-(D/E)xK superfamily relied on previous work (33).

### Identification of bacteroidetes genomes encoding T9SS

Taxonomy information was downloaded from the NCBI FTP server on December 20, 2019. Protein sequences of Bacteroidetes genomes (Taxonomy ID: 976) were retrieved from the local database. RPS-BLAST (22) was used to identify the T9SS core components in the Bacteroidetes genomes as well as the TANFOR-containing bacterial genomes. The proteins were aligned against five core proteins: GldK (TIGR03525), GldL (TIGR03513), GldM (TIGR03517), GldN (TIGR03523), and SprA (TIGR04189) (2, 18). Bacterial genomes encoding at least four of the five T9SS core components were regarded as harboring T9SS. The proteins in the TANFOR-containing genomes were also aligned against the T9SS-related proteins: SprB (pfam13573), type A CTD (TIGR04183), and type B CTD (TIGR04131) (Supplemental Data 3).

### Identification of domains enriched in TANFOR genomic neighborhoods

The protein sequences of genes located within 10 upstream/downstream genes of each TANFOR-encoding gene were analyzed using RPS-BLAST to identify conserved domains. For each conserved domain that was identified, the occurrence of the domain in TANFOR-encoding gene neighborhoods was determined (in cases where the same domain appeared more than one time in a TANFOR-encoding gene neighborhood, it was counted one time). This analysis was repeated for a group of non-TANFOR-encoding genes that were randomly sampled from TANFOR-containing genomes. To identify conserved domains that were enriched in TANFOR genomic neighborhoods, a contingency table was generated for each conserved domain and *P*-value was calculated using a one-tailed Fisher's exact test (Supplemental Data 4). The CDD version 3.17 contained 52,910 models. Thus, to correct for multiple hypotheses, the Bonferroni correction was applied, yielding a significance level of $0.05/52,910 = 9.45 \times 10^{-7}$. Conserved domains below this significance level are listed in Table 1.

## Supplementary Information

## Acknowledgements

This project received funding from the European Research Council under the European Union's Horizon 2020 research and innovation program (Grant

agreement No 714224), and from the Israel Science Foundation (grant no 920/17) to D Salomon. We thank Aya Vituri of the Statistical Consulting Lab of the Department of Statistics and Operations Research of Tel Aviv University for assistance with statistical analysis.

## Author Contributions

B Jana: investigation, methodology, and writing—review and editing.
D Salomon: conceptualization, supervision, funding acquisition, investigation, methodology, and writing—original draft.
E Bosis: conceptualization, formal analysis, supervision, investigation, methodology, and writing—original draft.

## Conflict of Interest Statement

The authors declare that they have no conflict of interest.

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
