## [Reviewer comments · Life Science Alliance]

Life Science Alliance

A novel class of polymorphic toxins in Bacteroidetes

Biswanath Jana, Dor Salomon, and Eran Bosis
DOI: <https://doi.org/10.26508/lsa.201900631>

Corresponding author(s): Dor Salomon, Tel Aviv University and Eran Bosis, ORT Braude College of Engineering

Review Timeline:	Submission Date:	2019-12-19
	Editorial Decision:	2020-01-20
	Revision Received:	2020-02-23
	Editorial Decision:	2020-03-03
	Revision Received:	2020-03-03
	Accepted:	2020-03-04

Scientific Editor: Andrea Leibfried

Transaction Report:

January 20, 2020

Re: Life Science Alliance manuscript #LSA-2019-00631-T

Dr. Dor Salomon
Tel Aviv University
Clinical Microbiology and Immunology
Ramat Aviv
Sackler Faculty of Medicine
Tel Aviv 6997801
Israel

Dear Dr. Salomon,

Thank you for submitting your manuscript entitled "Novel family of polymorphic toxins in Bacteroidetes" to Life Science Alliance. The manuscript was assessed by expert reviewers, whose comments are appended to this letter.

As you will see, the reviewers appreciate your analyses and provide constructive input on how to further strengthen it. We would thus like to invite you to submit a revised version of your work to us. In addition to the control (rev#1) and more minor changes requested, we concluded that it is important to address rev#1 and #2's concern regarding the lack of support for T9SS-dependent transport to elevate the value of your manuscript and your conclusions to others.

Thank you for this interesting contribution to Life Science Alliance. We are looking forward to receiving your revised manuscript.

Sincerely,

B. MANUSCRIPT ORGANIZATION AND FORMATTING:

Reviewer #1 (Comments to the Authors (Required)):

Novel family of polymorphic toxins in Bacteroidetes

Summary

Overall the parts characterising TANFOR-containing proteins and the new BaCT domain are solid, although lacking a few controls at times (see below). There's a lot of bioinformatics where these reviewers cannot comment on the methodology, but the conclusions mostly seem reasonable. The proposed T9SS link is only weakly supported and should be downplayed or supported with further evidence such as experiments in T9SS- bugs.

Title

This should probably start with "A".

Introduction

Line 30: Not all effectors can be classed as toxins, things like TALEs manipulate the host but aren't strictly toxins by my understanding - rephrase.

Line 31: There are some fairly niche secretion systems out there - perhaps rephrase to allow for these "Nine major/widespread secretion systems".

Line 49: Not all T9SS substrates are large, although the average is probably larger than other secretion systems, rephrase.

Line 69: "Importantly" is superfluous here.

Line 73: No strong evidence is presented to support the idea that BaCT is definitely an antibacterial toxin, just that it's a DNase and toxic when expressed in *E. coli*. The bar for proof of a role as a toxin would need to involve demonstration of activity in an infectious setting.

Results

TANFOR-containing proteins are polymorphic toxins

Perhaps use TANFOR-domain as used throughout the rest of the text

Line 98: signal peptides suggest that the proteins are exported to the periplasm, not necessarily secreted from the cell.

BaCT is a novel DNase toxin

Line 103: should say "known downstream toxin domain", they found one previously-unknown toxin domain and there could be more in the domains they didn't investigate.

Line 111: "could encode a cognate immunity protein". This feels a bit circular. At this stage in the story they don't have reason to believe the downstream ORFs encode immunity proteins until they decide the BaCT domain is a toxin. Being attached to the TANFOR domain in the same way as the PoNe domain seems enough reason to hypothesise it's a toxin and then hypothesise that the downstream ORF is an immunity protein on that basis.

Line 126: They should describe the divergences here, at least briefly.

Line 135: The facts the mutants are expressed shouldn't be notable, it's necessary to support their claim that these residues are part of the active site.

Figure 2C: it seems unnecessary to show such a long stretch when they only focus on two residues in the middle. This figure could also highlight the points of divergence from the PD-(D/E)xK superfamily.

Figure 2G: This blot should have the WT shown as well. It may be that the mutants are expressed to a much lesser extent than the WT. The lower bands don't appear in the pEmpty lane so it should be checked if they are more abundant in the mutants than in WT due to misfolding/degradation.

The protein encoded downstream of BaCT is its cognate immunity

Should add "protein" at the end of this subheading.

Line 153: should probably mention/cite CLANS here, not just in the methods.

Figure 3B: It would be clearer if the Flag and Myc blots were swapped so they're in the same order as the labelling above. It also looks like there might be a slight shift in the output lane of the Flag blot compared to the input.

Figure 3C: I don't think this panel really conveys any information without labels. The figures in the reference for CLANS look similar (without labelling). Could they use a different method or quote some statistic instead, even if it's just what proportion of sequences could be clustered?

TANFOR-toxins are genetically associated with T9SS

They do show a genetic link but they don't come up with any experimental/analysis evidence to support the claim that the toxins are transported by T9SS.

Line 174: "delivery" implies a directionality that T9SS secretion doesn't really have. We are not aware of any evidence that T9SS can target substrates to another cell.

Discussion

The first two paragraphs of the discussion talk about the T9SS link which came last in their results and also has the least data to back it up. These should at least come later in the discussion and be rather more hedged with caveats than at present.

Line 177: surely the TANFOR domain-containing proteins carry the toxin domains, not the TANFOR domain itself.

Line 189: this reasoning could be checked by looking for other two-step secretion systems in TANFOR domain-carrying genomes.

Line 191: delete thus.

Line 210: "immunity protein".

Line 235: the secreted proteins which mediate gliding motility are all adhesins as far as I'm aware.

Line 236: This sentence should mention the proximity to SprB repeat-containing proteins.

The TANFOR domains being upstream of cognate immunity proteins doesn't by itself suggest T9SS is involved in antibacterial activities.

Line 238: the most obvious way to test this is to examine the secretome of WT vs. T9SS- strains. T9SS- strains of *Tannerella forsythia* - for which the TANFOR domain is named - have been published. This would seem the key experiment to do with the authors wish to firmly suggest the link to T9SSs.

Overall the genetic link between TANFOR domains and T9SS is there and is suggestive that TANFOR helps with secretion of these toxin proteins. However, the authors spend an awful lot of time discussing this for the limited amount of evidence they actually have.

Methods

Strains and media

Line 253: should say "or 30 µg/ml kanamycin", currently it implies the authors added all three antibiotics at once.

Protein expression in *E. coli*

As well as not showing the WT (Figure 2G), they don't actually show that the mutant proteins are soluble and not misfolded in inclusion bodies.

In vitro DNase assays

The authors don't formally confirm that the protein was refolded here, although DNase activity suggests it was.

A gel of the purification to confirm there weren't any potentially active contaminants/different contaminants between the WT and mutant protein should be shown, probably in the SI. They did at least make sure to use the same amounts of protein between variants in this assay, though they don't comment on the relative yields.

In vivo DNase assays

This could easily be quantified by reporting the concentration of DNA in the genomic DNA elutions. This would show if the mutant variants are totally dead or have some small residual activity.

Clustering of proteins encoded downstream of BaCT using CLANS

The citation for CLANS should be included in the main text as well as here. The authors only name the software in the Figure 3 legend, not the main text of the results.

Identification of TANFOR- and BaCT-containing proteins

Line 423: Given they cite a specific program for identifying conserved domains (ref. 39 CDD/SPARCLE), they should probably name it here.

Reviewer #2 (Comments to the Authors (Required)):

This paper bioinformatically identifies a family of multidomain, periplasmically-targeted, Bacteroidetes proteins that minimally comprise an N-terminal conserved domain of unknown function (termed a TANFOR domain) and a toxin domain. Given this general structural arrangement for TANFOR-containing proteins, the authors then infer that a common TANFOR-associated domain, termed BacT, should be a toxin. They use in vitro and heterologous in vivo experiments to confirm that the BacT domain is toxic and to demonstrate that it has DNase activity. Finally, the authors go on to propose that the TANFOR domain-containing proteins are toxins exported across the outer membrane by the Bacteroidetes-specific Type IX Secretion System (T9SS).

The definition of a TANFOR-toxin family is interesting and the experimental analysis of the BacT domain sound. The identification of the TANFOR-containing proteins as T9SS substrates is both speculative and unexpected (and thus quite exciting if substantiated).

The arguments in favour of Type 9 targeting are as follows. TANFOR-toxin proteins will need to leave the producing cell to have an effect on their target cells; they can be inferred to be targeted to the periplasm by N-terminal Sec signal peptides and so, by inference, they need to be transported across the outer membrane of the producing cell; Type 9 is the only characterized, widely distributed outer membrane transport system in Bacteroidetes; strains encoding TANFOR-toxin proteins also normally (>95%) have a T9SS. Nevertheless, all known T9SS substrate proteins contain a defined C-terminal targeting domain (CTD) and the TANFOR-toxin proteins do not have such a domain. Thus it is a big leap to conclude that the TANFOR-toxin proteins are Type 9 secreted. To make this unexpected claim the authors really need to provide some experimental data. Specifically, they minimally need to show that a TANFOR-toxin protein is secreted and that this secretion is T9SS-dependent (gene knockouts in Flavobacteria and Cytophaga are generally quite straightforward using the plasmids developed by Mark McBride, and the T9SS is normally not an essential pathway for laboratory growth). If the authors are not going to provide such evidence then they need to considerably tone down their claims about T9SS targeting. In particular the claim that the TANFOR-toxins have been shown to be 'genetically associated' or 'genetically linked' with the T9SS is going too far. These terms imply either experimental evidence or physical linkage to the transport system genes, neither of which is true in this case. And dedicating the majority of the discussion to a process that we do not actually know is happening seems excessive.

In the absence of experimental evidence of T9SS targeting the level of significance of this paper would be around that of PLOS One or similar journals.

Other comments

I would like to see a few sentences in the introduction on what is known about secreted polymorphic toxins in general.

The results of the bioinformatics analysis is often described in qualitative terms when it would be possible, and more helpful, to have precise descriptions of what has been done or observed. For example, in the Fig. 1 legend we are told that genes are 'near' a downstream encoded toxin - what exactly is the 'nearness' criterion? On Pg4 'many' TANFOR-containing proteins do not have a toxin gene - so how many? On Pg9 TANFOR-encoding genes are 'near' a gene with a T9SS CTD - what is the 'nearness' criterion? Some of this is in the methods but the reader should not be asked to look through this when it is straightforward to specify these things in the main text.

In Fig.1 the legend needs to list what all the domain abbreviations are. The legend title makes it seem that all TANFOR containing proteins carry toxins but this is misleading as that there are other (in fact apparently `many') proteins with TANFOR domains but no toxins that are not shown. It's not very obvious to the reader that the domain architectures in Fig.1 are symbolic and not scaled and you have to read the legend to know that other domains have been removed from the figure. Given this, would it be better to join the two domains shown with a dashed line to give some indication that something else might be present between the two domains shown?

Pg4. `Many TANFOR-containing proteins did not carry ... an encoded toxin domain'. Since these are just bioinformatics predictions and the authors go on to show that some have a novel toxin domain the text should be reworded along the lines `Many TANFOR-containing proteins did not carry an identifiable/predicted... toxin domain'.

Is the proposed structural similarity of the BaCT domains to PD-(D/E)xK phosphodiesterases supported by fold prediction proteins such as PHYRE?

In the Fig.2 legend under C it should say that the cyan ovals depict PREDICTED/PROPOSED catalytic residues. What does `visualized as trihalo compounds' fluorescence of the immunoblot membrane' mean?

In Fig.2G why are there two bands identified by the Myc antibodies and does this matter? Are the proteins used for the in vivo DNA degradation experiment in Fig.2E the same Myc-tagged construct used to control for expression in Fig. 2G? And are these the identical samples analysed in two different ways in the two figures? If not, why not?

Why is the E. coli genomic DNA in Fig.2 a single band not much larger than 10kbp?

Pg 9 Figures are given for the co-occurrence of TANFOR-toxin protein genes and the T9SS. For comparison of the significance of this perhaps the authors could also give the level of co-occurrence of the toxins with the T6SS.

The authors assume that TANFOR domains are associated with Type 9 export. But the toxins have to be delivered into the cytoplasm (in most cases) of the target cell. So aren't the TANFOR domains as likely to be involved in toxin delivery rather than export?

Reviewer #3 (Comments to the Authors (Required)):

In this paper, Jana et al. propose that proteins containing a poorly characterized protein domain known as TANFOR are in fact substrates of the bacteria type 9 secretion system (T9SS). This proposition is supported by strong bioinformatics data and sets the stage for future experimental work to determine if TANFOR-domain containing proteins are (1) substrates of the T9SS and (2) function to mediate interbacterial competition. This work is fairly limited in terms of its experimental scope, but the implications are potentially profound because the authors may have identified a new mechanism for interbacterial antagonism. Below are minor comments for the authors consideration.

Abstract

- consider defining "TANFOR" in abstract.

Introduction

Lines 42 - 44 : "The association of effectors with a specific secretion system is mediated by a delivery domain or a short terminal peptide recognized by the secretion machinery."

- This is a generalization that isn't necessarily true for all secretion systems. For example, for the bacterial type VI secretion system, the picture is more complicated because many effectors don't have a delivery domain or terminal peptide. This could be expanded by mentioning the existence of chaperones that mediate effector delivery to specific secretion systems as well, or just say "many systems/with exceptions."

Results

Section 1: TANFOR-containing proteins are polymorphic toxins

- It would be very helpful to know the distribution of these proteins within the Bacteroidetes phylum. Are these putative toxins found in only a few species or widespread? Are there multiple TANFOR containing proteins encoded within a single genome? Do you see putative eukaryotic and bacterial targeting toxins in the same species? Gene and/or species tree might help communicate this effectively.

- "# of occurrences" in figure 1 is confusing. What is meant by this? The # of occurrences within the discovered 339 TANFOR-containing proteins?

Section 2: BaCT is a novel DNase toxin

Lines 103 - 105: "Many TANFOR-containing proteins did not carry or have a downstream encoded toxin domain. Therefore, we set out to determine whether the C-termini of TANFOR-containing proteins with no defined activity are novel toxin domains."

- The way this is written comes across as there being no C-terminal domain whatsoever, but what I think is meant is that there is no identifiable domain family within the C-terminus of the TANFOR-containing proteins.

- "# of occurrences" - again not sure what this is in reference to.

- Comparison of the predicted BaCT domain active site to that of the PD-(D/E)xK phosphodiesterase superfamily could be better illustrated with an alignment to canonical PD-(D/E)xK phosphodiesterase proteins. An alignment would also help the argument that these are a distinct/distantly related family of DNases. Minimally, please include the sequence for the phosphodiesterase superfamily active sites.

- Should all instances of Latin be italicized?

- Typo in figure 2G - should read "E1724A" not "11724A"

Section 3: The protein encoded downstream of BaCT is its cognate immunity

- The authors argue that WP_081912444.1 is an antibacterial toxin based on the presence of a downstream gene that encodes an immunity protein that negates WP_081912444.1 toxicity. However, given the proposed function of this toxin is to degrade DNA non-specifically (based on the DNase assay) - it is possible that WP_081912444.1 is active against Eukaryotic cells as well.

Perhaps it is worth testing if this protein is toxic towards eukaryotic cells since there is precedent for dual targeting effectors in other secretion systems?

- Figure 3C: This figure panel has no context. What do the axes mean? Labeling a few of the spots might help - show us the spot that represents the immunity you have tested so we know where to look for the grouping. Show controls that are clearly not part of the family on the outside of the clustering.

Section 4: TANFOR-toxins are genetically associated with the T9SS

- Would be nice to know more about the genomes that lack a T9SS but contain TANFOR genes. What do the authors propose these TANFOR proteins do?

Reviewer #1 (Comments to the Authors (Required)):

Novel family of polymorphic toxins in Bacteroidetes

Summary

Overall the parts characterising TANFOR-containing proteins and the new BACT domain are solid, although lacking a few controls at times (see below). There's a lot of bioinformatics where these reviewers cannot comment on the methodology, but the conclusions mostly seem reasonable. The proposed T9SS link is only weakly supported and should be downplayed or supported with further evidence such as experiments in T9SS- bugs.

Title

This should probably start with "A".

Added

Introduction

Line 30: Not all effectors can be classed as toxins, things like TALEs manipulate the host but aren't strictly toxins by my understanding - rephrase.

The sentence has been rephrased.

Line 31: There are some fairly niche secretion systems out there - perhaps rephrase to allow for these "Nine major/widespread secretion systems".

We added the word "major" as suggested.

Line 49: Not all T9SS substrates are large, although the average is probably larger than other secretion systems, rephrase.

We removed the word "large".

Line 69: "Importantly" is superfluous here.

We removed the word "Importantly".

Line 73: No strong evidence is presented to support the idea that BaCT is definitely an antibacterial toxin, just that it's a DNase and toxic when expressed in E. coli. The bar for proof of a role as a toxin would need to involve demonstration of activity in an infectious setting.

The term "toxin" is used to describe proteins that possess activities deleterious to the cell, and it does not necessarily imply an inter-cellular context. For example, bacteria possess toxin-antitoxin modules that regulate bacterial growth intracellularly and not in an infectious setting. In this work, we demonstrate that BaCT and BaCTic function in a similar manner, and we therefore refer to BaCT as a toxin.

Results

TANFOR-containing proteins are polymorphic toxins

Perhaps use TANFOR-domain as used throughout the rest of the text

The subheading was changed to "Proteins containing a TANFOR domain are polymorphic toxins".

Line 98: signal peptides suggest that the proteins are exported to the periplasm, not necessarily secreted from the cell.

We agree with the reviewer and we have corrected the sentence accordingly to "...suggesting that TANFOR-containing proteins are exported to the periplasm".

BaCT is a novel DNase toxin

Line 103: should say "known downstream toxin domain", they found one previously-unknown toxin domain and there could be more in the domains they didn't investigate.

The sentence has been rephrased to say: "Many TANFOR-containing proteins did not contain a predicted toxin domain and did not have an identifiable toxin encoded downstream".

Line 111: "could encode a cognate immunity protein". This feels a bit circular. At this stage in the story they don't have reason to believe the downstream ORFs encode immunity proteins until they decide the BaCT domain is a toxin. Being attached to the TANFOR domain in the same way as the PoNe domain seems enough reason to hypothesize it's a toxin and then hypothesize that the downstream ORF is an immunity protein on that basis.

We removed the first part of the sentence, and it now reads: "We hypothesized that BaCT functions as an antibacterial toxin domain".

Line 126: They should describe the divergences here, at least briefly.

The discussion of divergence was removed from the text, and we have added supplementary Fig. S1 showing the similarity between the canonical conserved motifs of the PD-(D/E)xK superfamily and those of BaCT. A multiple sequence alignment of BaCT family members is also provided.

Line 135: The facts the mutants are expressed shouldn't be notable, it's necessary to support their claim that these residues are part of the active site.

The sentence was rephrased, and now reads: "The mutant forms of BaCT were expressed, as evident by immunoblotting (Fig. 2G), demonstrating that the loss of toxicity did not result from impaired expression of the mutants".

Figure 2C: it seems unnecessary to show such a long stretch when they only focus on two residues in the middle. This figure could also highlight the points of divergence from the PD-(D/E)xK superfamily.

We prefer not to change this panel since we believe that readers will get a better sense of BaCT if we provide a longer stretch of residues, even if most are not well conserved. Please see our reply above regarding revising the divergence discussion and addition of comparison to PD-(D/E)xK superfamily motifs in supplementary Fig. S1.

Figure 2G: This blot should have the WT shown as well. It may be that the mutants are expressed to a much lesser extent than the WT. The lower bands don't appear in the pEmpty lane so it should be checked if they are more abundant in the mutants than in WT due to misfolding/degradation.

We could not detect the WT BaCT expression, possibly due to it being extremely toxic. We observed the same phenomenon in the past with other bacterial toxins. However, we do show its expression compared to that of the mutants in the presence of the immunity protein (Fig. 3B), and we now also provide a stained gel (Supplementary Fig. S2) showing that both the WT and a mutant BaCT that were purified and used in the in vitro DNase assay (Fig. 2D) have similar degradation patterns.

The protein encoded downstream of BaCT is its cognate immunity

Should add "protein" at the end of this subheading.

The subheading was rephrased, and now reads: "The gene downstream of BaCT encodes its cognate immunity protein".

Line 153: should probably mention/cite CLANS here, not just in the methods.

The citation was added.

Figure 3B: It would be clearer if the Flag and Myc blots were swapped so they're in the same order as the labelling above. It also looks like there might be a slight shift in the output lane of the Flag blot compared to the input.

Please see the revised Fig. 3B, in which we also include the BaCT mutants.

Figure 3C: I don't think this panel really conveys any information without labels. The figures in the reference for CLANS look similar (without labelling). Could they use a different method or quote some statistic instead, even if it's just what proportion of sequences could be clustered?

Please see the revised Fig. 3C, in which included a title for the panel and highlighted the experimentally tested BaCTic in red. We have also added the numbers of clustered BaCTic sequences (57/65, 87.7%) in the text.

TANFOR-toxins are genetically associated with T9SS

They do show a genetic link but they don't come up with any experimental/analysis evidence to support the claim that the toxins are transported by T9SS.

In the revised manuscript, we now provide an unbiased analysis of domains enriched in the neighborhoods of TANFOR-encoding genes (Table 1 and Supplementary Dataset S4). We identified 7 enriched domains out of 52,910 available models in the Conserved Domain Database. Out of the 7 enriched domains, 2 (SprB and Por_Secre_tail) are directly associated with T9SS substrates. However, since we cannot provide an experimental validation demonstrating T9SS-mediated TANFOR secretion, we have toned down this aspect of the work throughout the manuscript.

Line 174: "delivery" implies a directionality that T9SS secretion doesn't really have. We are not aware of any evidence that T9SS can target substrates to another cell.

We rephrased "delivered" to "exported".

Discussion

The first two paragraphs of the discussion talk about the T9SS link which came last in their results and also has the least data to back it up. These should at least come later in the discussion and be rather more hedged with caveats than at present.

In the revised version of this manuscript, we have toned down the discussion regarding the possibility of T9SS-mediated export of TANFOR-containing proteins.

Line 177: surely the TANFOR domain-containing proteins carry the toxin domains, not the TANFOR domain itself.

Please see the revised discussion.

Line 189: this reasoning could be checked by looking for other two-step secretion systems in TANFOR domain-carrying genomes.

We now provide an unbiased analysis of domains enriched in TANFOR neighboring genes, in which we found that TANFOR-encoding genes are genetically associated with T9SS secreted components, but not with components of other secretion systems (Supplementary Dataset S4).

Line 191: delete thus.

Done

Line 210: "immunity protein".

Added.

Line 235: the secreted proteins which mediate gliding motility are all adhesins as far as I'm aware.

This has been resolved in the revised Results and Discussion section.

Line 236: This sentence should mention the proximity to SprB repeat-containing proteins. The TANFOR domains being upstream of cognate immunity proteins doesn't by itself suggest T9SS is involved in antibacterial activities.

This hypothetical parts regarding the link between TANFOR and T9SS have been extensively modified in this revised version. As mentioned above, we substituted the proximity analysis with an unbiased domain-enrichment analysis (supplementary Dataset S4 and Table 1).

Line 238: the most obvious way to test this is to examine the secretome of WT vs. T9SS- strains. T9SS- strains of *Tannerella forsythia* - for which the TANFOR domain is named - have been published. This would seem the key experiment to do with the authors wish to firmly suggest the link to T9SSs.

We agree with the reviewer that such an experiment is required to firmly establish that TANFOR-containing proteins are exported by T9SS. However, since we presently lack expertise to establish this experimental system in our labs, we will pursue this in the future through

possible collaborations.

Overall the genetic link between TANFOR domains and T9SS is there and is suggestive that TANFOR helps with secretion of these toxin proteins. However, the authors spend an awful lot of time discussing this for the limited amount of evidence they actually have.

As mentioned above, we have toned down the suggested link between TANFOR and T9SS-mediated export in this revised version of the manuscript.

Methods

Strains and media

Line 253: should say "or 30 µg/ml kanamycin", currently it implies the authors added all three antibiotics at once.

Corrected.

Protein expression in E. coli

As well as not showing the WT (Figure 2G), they don't actually show that the mutant proteins are soluble and not misfolded in inclusion bodies.

We were able to purify BaCT variants in the pull down assays and for the in vitro DNase assay, demonstrating that at least part of the expressed protein population is soluble. The pull down assay demonstrating interaction with BaCTic, as well as the in vitro DNase activity of WT BaCT, indicate that the proteins are properly folded.

In vitro DNase assays

The authors don't formally confirm that the protein was refolded here, although DNase activity suggests it was.

The DNase activity indicates proper refolding.

A gel of the purification to confirm there weren't any potentially active contaminants/different contaminants between the WT and mutant protein should be shown, probably in the SI. They did at least make sure to use the same amounts of protein between variants in this assay, though they don't comment on the relative yields.

We now provide a gel showing the purified BaCT variants used in the in vitro DNase assay (Supplementary Fig. S2).

In vivo DNase assays

This could easily be quantified by reporting the concentration of DNA in the genomic DNA elutions. This would show if the mutant variants are totally dead or have some small residual activity.

Quantification of DNA in these assays will not be reliable due to inaccuracy of nanodrop measurements, presence of remaining RNA that can affect results, and because measuring DNA concentration cannot differentiate properly between degraded and complete genome forms. We also note that the results shown in Fig. 2 were meant to be assessed together and are complementary. When taken together, the results demonstrating loss of toxicity when measuring the effect of BaCT mutants on bacterial growth, and loss of SOS response activation, support the conclusion that the point mutants lost the DNase activity. Moreover, even if some minor residual activity remains in the point mutants, it does not change any of the conclusions in our work.

Clustering of proteins encoded downstream of BaCT using CLANS

The citation for CLANS should be included in the main text as well as here. The authors only name the software in the Figure 3 legend, not the main text of the results.

We now specify the use of CLANS in the main text.

Identification of TANFOR- and BaCT-containing proteins

Line 423: Given they cite a specific program for identifying conserved domains (ref. 39 CDD/SPARCLE), they should probably name it here.

The name was added.

Reviewer #2 (Comments to the Authors (Required)):

This paper bioinformatically identifies a family of multidomain, periplasmically-targeted, Bacteroidetes proteins that minimally comprise an N-terminal conserved domain of unknown function (termed a TANFOR domain) and a toxin domain. Given this general structural arrangement for TANFOR-containing proteins, the authors then infer that a common TANFOR-associated domain, termed BacT, should be a toxin. They use in vitro and heterologous in vivo experiments to confirm that the BacT domain is toxic and to demonstrate that it has DNase activity. Finally, the authors go on to propose that the TANFOR domain-containing proteins are toxins exported across the outer membrane by the Bacteroidetes-specific Type IX Secretion System (T9SS).

The definition of a TANFOR-toxin family is interesting and the experimental analysis of the BacT domain sound. The identification of the TANFOR-containing proteins as T9SS substrates is both speculative and unexpected (and thus quite exciting if substantiated).

The arguments in favour of Type 9 targeting are as follows. TANFOR-toxin proteins will need to leave the producing cell to have an effect on their target cells; they can be inferred to be targeted to the periplasm by N-terminal Sec signal peptides and so, by inference, they need to be transported across the outer membrane of the producing cell; Type 9 is the only characterized, widely distributed outer membrane transport system in Bacteroidetes; strains encoding TANFOR-toxin proteins also normally (>95%) have a T9SS. Nevertheless, all known T9SS substrate proteins contain a defined C-terminal targeting domain (CTD) and the TANFOR-toxin proteins do not have such a domain. Thus it is a big leap to conclude that the TANFOR-toxin proteins are Type 9 secreted. To make this unexpected claim the authors really need to provide some experimental data. Specifically, they minimally need to show that a TANFOR-toxin protein is secreted and that this secretion is T9SS-dependent (gene knockouts in Flavobacteria and Cytophaga are generally quite straightforward using the plasmids developed by Mark McBride, and the T9SS is normally not an essential pathway for laboratory growth). If the authors are not going to provide such evidence then they need to considerably tone down their claims about T9SS targeting.

In particular the claim that the TANFOR-toxins have been shown to be 'genetically associated' or 'genetically linked' with the T9SS is going too far. These terms imply either experimental evidence or physical linkage to the transport system genes, neither of which is true in this case. And dedicating the majority of the discussion to a process that we do not actually know is happening seems excessive.

In the absence of experimental evidence of T9SS targeting the level of significance of this paper would be around that of PLOS One or similar journals.

In the revised version of the manuscript, we provide additional, unbiased computational data to support the genetic association between TANFOR and T9SS substrates. We find that genes neighboring TANFOR-encoding genes are significantly enriched with SprB and Por_Secret_tail-encoding genes, but not with other secretion systems-related genes (Table 1 and Supplementary Dataset S4). Nevertheless, since we are currently unable to provide supporting experimental data, we toned down this aspect of the manuscript.

Other comments

I would like to see a few sentences in the introduction on what is known about secreted polymorphic toxins in general.

This was added in the revised Introduction section.

The results of the bioinformatics analysis is often described in qualitative terms when it would be possible, and more helpful, to have precise descriptions of what has been done or observed. For example, in the Fig. 1 legend we are told that genes are 'near' a downstream encoded toxin - what exactly is the 'nearness' criterion? On Pg4 'many' TANFOR-containing proteins do not have a toxin gene - so how many? On Pg9 TANFOR-encoding genes are 'near' a gene with a T9SS CTD - what is the 'nearness' criterion? Some of this is in the methods but the reader should not be asked to look through this when it is straightforward to specify these things in the main text.

In the revised manuscript, we provide more accurate description of these terms in the text and in the figure legends.

In Fig.1 the legend needs to list what all the domain abbreviations are. The legend title makes it seem that all TANFOR containing proteins carry toxins but this is misleading as that there are other (in fact apparently 'many') proteins with TANFOR domains but no toxins that are not shown. It's not very obvious to the reader that the domain architectures in Fig.1 are symbolic and not scaled and you have to read the legend to know that other domains have been removed from the figure. Given this, would it be better to join the two domains shown with a dashed line to give some indication that something else might be present between the two domains shown?

Please see revised Fig. 1 and its legend, in which we have addressed the reviewer's suggestions.

Pg4. 'Many TANFOR-containing proteins did not carry ... an encoded toxin domain'. Since these are just bioinformatics predictions and the authors go on to show that some have a novel toxin domain the text should be reworded along the lines 'Many TANFOR-containing proteins did not carry an identifiable/predicted... toxin domain'.

The sentence was rephrased, and now reads: "Many TANFOR-containing proteins did not contain a predicted toxin domain and did not have an identifiable toxin encoded downstream".

Is the proposed structural similarity of the BaCT domains to PD-(D/E)xK phosphodiesterases supported by fold prediction proteins such as PHYRE?

Members of the PD-(D/E)xK superfamily exhibit extreme sequence and structure diversity (Knizewski et al, BMC Struct Biol, 2007; Steczkiewicz et al, NAR, 2012), making fold predictions unreliable. To provide a better comparison to PD-(D/E)xK members, we now include Supplementary Fig. S1 that shows the canonical PD-(D/E)xK compared with the secondary structure prediction and the multiple sequence alignment of the BaCT family.

In the Fig.2 legend under C it should say that the cyan ovals depict PREDICTED/PROPOSED catalytic residues.

The legend was modified, and the sentence now reads: "Cyan ovals denote conserved predicted catalytic residues".

What does 'visualized as trihalo compounds' fluorescence of the immunoblot membrane' mean?

This refers to a commercial Stain-free technology provided by Bio-Rad in their "stain-free" gels. A trihalo compound found in the gels is covalently bound to tryptophan residues in resolved proteins, allowing immediate visualization of proteins in the gel or after transfer to membranes following a short photoactivation. We revised the relevant sentence in the legend to provide more details to the readers: "Loading control (LC), visualized as trihalo compounds' fluorescence of the immunoblot membrane (Stain-Free imaging technology; Bio-Rad), is shown for total protein lysates".

In Fig.2G why are there two bands identified by the Myc antibodies and does this matter? Are the proteins used for the in vivo DNA degradation experiment in Fig.2E the same Myc-tagged construct used to control for expression in Fig. 2G? And are these the identical samples analysed in two different ways in the two figures? If not, why not?

The lower band detected in the immunoblot shown in Fig. 2G is probably a degradation product of the toxin domain. This lower band were also detected when the WT BaCT was expressed, as seen in Fig. 3B.

The same Myc-His-tagged constructs were used for all experiments shown in panels A, D, E, F, and G.

The protein expression shown in panel G was produced from collecting the induced *E. coli* cultures at the end of the growth experiment whose results are shown in panel A (as described in the "Protein expression in *E. coli*" Methods section). This was done to ensure that the lack of deleterious effect on bacterial growth in cultures containing plasmids for expression of the mutant BaCT variants did not result from lack of protein expression.

Why is the *E. coli* genomic DNA in Fig.2 a single band not much larger than 10kbp?

This is the standard appearance of gDNA resolved on an agarose gel (for reference, please see sample gel images produced using various commercial kits:

<https://www.thermofisher.com/order/catalog/product/K0721#/K0721>,
<https://www.biocat.com/products/17900-NB>).

Pg 9 Figures are given for the co-occurrence of TANFOR-toxin protein genes and the T9SS. For comparison of the significance of this perhaps the authors could also give the level of co-occurrence of the toxins with the T6SS.

In the revised manuscript, we now provide an unbiased approach to identify domains that are enriched in TANFOR genome neighborhoods. We identified 7 enriched domains out of 52,910 available models in the Conserved Domain Database. Of the 7 enriched domains, the only domains that are associated with a secretion system are SprB and Por secretion system C-terminal sorting domain, which are found in T9SS substrates (Table 1 and new Supplementary Dataset S4).

The authors assume that TANFOR domains are associated with Type 9 export. But the toxins have to be delivered into the cytoplasm (in most cases) of the target cell. So aren't the TANFOR domains as likely to be involved in toxin delivery rather than export?

We agree with the reviewer's comment. However, please note that this part was removed from the text as we toned down the hypothetical parts on the possible link of TANFOR and T9SS-mediated export.

Reviewer #3 (Comments to the Authors (Required)):

In this paper, Jana et al. propose that proteins containing a poorly characterized protein domain known as TANFOR are in fact substrates of the bacteria type 9 secretion system (T9SS). This proposition is supported by strong bioinformatics data and sets the stage for future experimental work to determine if TANFOR-domain containing proteins are (1) substrates of the T9SS and (2) function to mediate interbacterial competition. This work is fairly limited in terms of its experimental scope, but the implications are potentially profound because the authors may have identified a new mechanism for interbacterial antagonism. Below are minor comments for the authors consideration.

Abstract

- consider defining "TANFOR" in abstract.

A short description was added to the abstract.

Introduction

Lines 42 - 44 : "The association of effectors with a specific secretion system is mediated by a delivery domain or a short terminal peptide recognized by the secretion machinery."

- This is a generalization that isn't necessarily true for all secretion systems. For example, for the bacterial type VI secretion system, the picture is more complicated because many effectors don't have a delivery domain or terminal peptide. This could be expanded by mentioning the existence of chaperones that mediate effector delivery to specific secretion systems as well, or just say "many systems/with exceptions."

This section has been rephrased in the revised manuscript.

Results

Section 1: TANFOR-containing proteins are polymorphic toxins

- It would be very helpful to know the distribution of these proteins within the Bacteroidetes phylum. Are these putative toxins found in only a few species or widespread? Are there multiple TANFOR containing proteins encoded within a single genome? Do you see putative eukaryotic and bacterial targeting toxins in the same species? Gene and/or species tree might help communicate this effectively.

In the revised manuscript, we now include an analysis of the presence and number of TANFOR-encoding genes in Bacteroidetes genomes available in RefSeq (supplementary Dataset S2, ~5% of Bacteroidetes genomes encode 1-5 TANFOR-containing proteins). This dataset also includes an analysis of the presence of T9SS in each Bacteroidetes genome. The results are summarized and described in the text.

- "# of occurrences" in figure 1 is confusing. What is meant by this? The # of occurrences within the discovered 339 TANFOR-containing proteins?

'# of occurrences' refers to the number of TANFOR-containing proteins that were found to have the indicated domain architecture. This is now explained in the Figure legend.

Section 2: BaCT is a novel DNase toxin

Lines 103 - 105: "Many TANFOR-containing proteins did not carry or have a downstream encoded toxin domain. Therefore, we set out to determine whether the C-termini of TANFOR-containing proteins with no defined activity are novel toxin domains."

- The way this is written comes across as there being no C-terminal domain whatsoever, but what I think is meant is that there is no identifiable domain family within the C-terminus of the TANFOR-containing proteins.

This section has been rephrased in the revised manuscript.

- "# of occurrences" - again not sure what this is in reference to.

'# of occurrences' refers to the number of BaCT-containing proteins that were found to have the indicated domain architecture. This is now explained in the Figure legend.

- Comparison of the predicted BaCT domain active site to that of the PD-(D/E)xK phosphodiesterase superfamily could be better illustrated with an alignment to canonical PD-(D/E)xK phosphodiesterase proteins. An alignment would also help the argument that these are a distinct/distantly related family of DNases. Minimally, please include the sequence for the phosphodiesterase superfamily active sites.

We now include Supplementary Fig. S1 that shows the canonical PD-(D/E)xK compared with the secondary structure prediction and the multiple sequence alignment of the BaCT family. As shown in supplementary Fig. S1, the conserved motifs shown at the top are mostly conserved also in BaCT. Please note that members of the PD-(D/E)xK superfamily exhibit extreme sequence and structure diversity (Knizewski et al, BMC Struct Biol, 2007; Steczkiewicz et al, NAR, 2012), making multiple sequence alignment of one family (e.g. BaCT) with other families unreliable.

- Should all instances of Latin be italicized?

Examining recent publication in this journal suggests that terms such as "in vivo" and "in vitro" are not italicized, but of course we will adhere to the journal's guidelines as they are indicated by the editor.

- Typo in figure 2G - should read "E1724A" not "11724A"

Corrected.

Section 3: The protein encoded downstream of BaCT is its cognate immunity

- The authors argue that WP_081912444.1 is an antibacterial toxin based on the presence of a downstream gene that encodes an immunity protein that negates WP_081912444.1 toxicity. However, given the proposed function of this toxin is to degrade DNA non-specifically (based on

the DNase assay) - it is possible that WP_081912444.1 is active against Eukaryotic cells as well. Perhaps it is worth testing if this protein is toxic towards eukaryotic cells since there is precedent for dual targeting effectors in other secretion systems?

Indeed, we cannot rule out the possibility that TANFOR-containing toxins, and specifically BaCT, function as trans-kingdom toxins. However, since we currently do not have a working model system to test possible contributions of these toxins in either interbacterial competition or anti-eukaryotic toxicity, we believe that this remains beyond the scope of the current work.

- Figure 3C: This figure panel has no context. What do the axes mean? Labeling a few of the spots might help - show us the spot that represents the immunity you have tested so we know where to look for the grouping. Show controls that are clearly not part of the family on the outside of the clustering.

The panel in Fig. 3C depicts two-dimensional clustering of protein sequences using all-against-all pairwise alignment. This is now better explained in the Figure legend. Please note that the black lines are simply a frame, not axis. In addition, to clarify the results we have added a title to the panel, and we marked the experimentally validated BaCTic protein with a red circle.

We believe that adding non-related sequences to the analysis figure could potentially be confusing to the readers. Therefore, to demonstrate that the CLANS tools that we used can discriminate between protein families and cluster them separately, we provide here an analysis of two protein families. The first is BaCTic (red circles), and the second is PoNi (blue circles, a family of immunity proteins that antagonize a different PD-(D/E)xK phosphodiesterase DNase toxin family; Jana et al, Nature Communications, 2019). As seen in the figure below, the CLANS

BaCTic (used in Fig. 3C)
PoNi (Jana et al, Nat Comms, 2019)
tool grouped the two protein families into distinct clusters.

Section 4: TANFOR-toxins are genetically associated with the T9SS

- Would be nice to know more about the genomes that lack a T9SS but contain TANFOR genes. What do the authors propose these TANFOR proteins do?

The genomes that encode TANFOR but have no T9SS are proteobacterial genomes (e.g. *Pseudomonas* and *Halomonas*) and a Spirochaetia genome (*Leptospira*). This can be seen in supplementary Dataset S5. Examining the relevant TANFOR-containing proteins in these genomes reveals that they appear quite different from the majority of TANFOR-containing proteins that are found in Bacteroidetes genomes. These proteins are shorter (<600 amino acids), and the predicted TANFOR domain is often in a region that overlaps with other random conserved domains. We are not sure why these proteins passed our conservative E value cutoffs for TANFOR domains, but since they did we did not think that it is appropriate to manually remove them from our lists. It is our impression that these are not true TANFOR domains.

March 3, 2020

RE: Life Science Alliance Manuscript #LSA-2019-00631-TR

Dr. Dor Salomon
Tel Aviv University
Clinical Microbiology and Immunology
Ramat Aviv
Sackler Faculty of Medicine
Tel Aviv 6997801
Israel

Dear Dr. Salomon,

Thank you for submitting your revised manuscript entitled "A novel class of polymorphic toxins in Bacteroidetes".

As you will see, one of the reviewers thinks that the computational data provided to support the genetic association between TANFOR and T9SS substrates is not convincing. This reviewer also still thinks that the value provided to others is still mediocre at this stage.

We discussed your work in light of all input and concluded that given the two more supportive reviewers, we can publish your paper in Life Science Alliance despite the missing more definitive insight. Please log in one more time into our submission system to move all files to the final version of your manuscript and to fill in the electronic license to publish form.

A. FINAL FILES:

B. MANUSCRIPT ORGANIZATION AND FORMATTING:

Sincerely,

Reviewer #2 (Comments to the Authors (Required)):

The authors have satisfactorily dealt with the issues raised in my review. Note that in the case of the suggestion that the toxins are exported by the T9SS the authors have chosen to temper their claims rather than address the issue experimentally. That's OK, but as I indicated in my original review this produces a PLOS One-level paper. I also note that linkage between toxin genes and other substrates of the T9SS is unlikely to predict use of the T9SS since even components of the T9SS are not usually genetically linked.

Reviewer #3 (Comments to the Authors (Required)):

The authors have satisfactorily addressed my comments.

March 4, 2020

RE: Life Science Alliance Manuscript #LSA-2019-00631-TRR

Dr. Dor Salomon
Tel Aviv University
Clinical Microbiology and Immunology
Ramat Aviv
Sackler Faculty of Medicine
Tel Aviv 6997801
Israel

Dear Dr. Salomon,

Thank you for submitting your Research Article entitled "A novel class of polymorphic toxins in Bacteroidetes". It is a pleasure to let you know that your manuscript is now accepted for publication in Life Science Alliance. Congratulations on this interesting work.

DISTRIBUTION OF MATERIALS:

Again, congratulations on a very nice paper. I hope you found the review process to be constructive and are pleased with how the manuscript was handled editorially. We look forward to future exciting submissions from your lab.

Sincerely,

Andrea Leibfried, PhD
Executive Editor
Life Science Alliance
Meyrhofstr. 1
69117 Heidelberg, Germany
t +49 6221 8891 502
e a.leibfried@life-science-alliance.org
www.life-science-alliance.org